# Self-Supervised Synthetic Cerebral Vessel Tree Generation using Semantic Signed Distance Fields

**Thijs P. Kuipers**[1,2] (iD)                                      T.P.KUIPERS@AMSTERDAMUMC.NL
**Praneeta R. Konduri**[1,2]                                     P.R.KONDURI@AMSTERDAMUMC.NL
**Erik J. Bekkers**[3]                                                      E.J.BEKKERS@UVA.NL
**Henk A. Marquering**[1,2]                           H.A.MARQUERING@AMSTERDAMUMC.NL

[1] *Department of Biomedical Engineering and Physics, Amsterdam UMC, The Netherlands*

[2] *Department of Radiology and Nuclear Medicine, Amsterdam UMC, The Netherlands*

[3] *Informatics Institute, University of Amsterdam, The Netherlands*

**Editors:** Accepted for publication at MIDL 2025

## Abstract

Advances in in-silico clinical trails for the development of novel treatment and devices for acute ischemic stroke have driven the creation of synthetic virtual patient populations to address the lack of large real-world datasets. Recent work proposed a method for generating semantic vascular centerline tree of the major cerebral arteries using pointcloud diffusion. However, this approach relies on separate post-processing algorithms to reconstruct the vessel tree topology, which does not generalize well to more topologically complex trees. To overcome this limitation, we introduce semantic signed distance fields for modeling cerebral vessel trees in a fully self-supervised manner. Our approach bypasses the need for separate reconstruction of the tree topology, and can be trained directly on shape-surfaces. Our method combines a variational autoencoder for encoding shapes to robust latent shape representations with a latent-diffusion model for generating synthetic vessel trees. By generating surface geometry directly, our approach eliminates the need for post-processing steps, enabling the generation of high-quality and topologically complex cerebral vessel trees.

**Keywords:** Shape generation, implicit neural representation, self-supervision, in-silico clinical trials, latent diffusion, semantic vascular geometry.

## 1. Introduction

Advancements in computational modeling have enabled high-fidelity patient-specific treatment simulations for acute ischemic stroke (AIS) (Luraghi et al., 2021; Liu et al., 2022). These simulations support the promise of in-silico clinical trials (ISCTs) as alternatives to traditional trials for developing medical treatments and devices (Konduri et al., 2020; Miller et al., 2023). However, ISCTs require large virtual populations of high-quality 3D vascular geometries, which are challenging to create due to resource-intensive processes. Synthetic data generation, e.g., with deep generative models, addresses these limitations by generating diverse, high-quality synthetic geometries from limited real data. Because synthetic populations bypass privacy restrictions, they enable data sharing and support downstream tasks reliant on large datasets.

Several methods for generating 3D vascular geometry have been proposed in recent literature. Danu et al. (2019) utilize an image-based generative approach and represent

the geometry as discrete 3D voxel occupancy grids, limiting the resolution of the generated vessels. Wolterink et al. (2018) generate single-branch centerline graphs sequentially using a generative adversarial network (GAN), but do not support bifurcations. Expanding on this concept, Feldman et al. (2023) generate vessel centerline tree graphs using a recursive variational autoencoder that does supports bifurcations. However, the method is limited by its recursion depth and does not support looping vessels or multiple disconnected graphs, which all occur in cerebral vascular structures such as the circle of Willis. As such, these methods are not directly applicable for generating cerebral vessel trees. In contrast, Sinha and Hamarneh (2024) represent 3D vascular geometry using implicit neural representations (INRs) with occupancy fields, making this method free of topological restrictions.

Besides topological restrictions, a major limitation of the aforementioned methods is the lack of semantic labeling of the individual vessels in the generated trees, which is crucial in computational stroke treatment models for the automatic placement of thrombi in specific vessels and locations. Additionally, having access to semantic labels allows for more robust evaluation of the synthetic vessel trees by assessing the quality of each individual vessel within the tree. As a result, Kuipers et al. (2024) introduced a point cloud-based diffusion approach for generating semantic cerebral vessel trees. However, a separate rule-based post-processing algorithms is required to reconstruct the vessel tree topology for which we show that it does not generalize well to topologically complex trees.

In this work, we employ INRs and propose representing cerebral vessel trees as semantic signed distance fields (SDFs), avoiding the need for separate post-processing algorithms. SDFs represent the distance from a point to the surface of the shape, with the sign indicating whether the point is inside or outside the shape. INRs provide several advantages over voxel- or point cloud-based methods, including memory efficiency, support for arbitrary resolution, continuity, and automatic differentiation, e.g., for computing surface normals (Berzins et al., 2024). In the generative setting, INRs are typically optimized in a supervised manner using ground truth scalar fields, which require access to *watertight* geometry, i.e., closed surfaces representing a

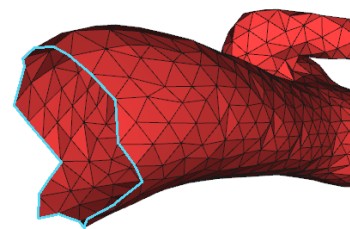

Figure 1: Non-closed mesh from (Yang et al., 2020).

volume (Chibane et al., 2020). However, watertight geometry is often unavailable, particularly for tubular vascular structures, as seen in Figure 1, or when the surface geometry is represented as a point cloud. Moreover, obtaining accurate watertight geometry often involves labor-intensive manual processing. Building on the approach of Alblas et al. (2022), we leverage the inductive bias of SDFs, i.e., SDFs satisfy the Eikonal equation, extending the fully self-supervised learning of implicit neural shapes to the generative setting. As a result, our model does not require access to ground truth signed distances or occupancy grids, making it compatible with any type of surface representation, and will always yield a mesh that is watertight.

Our generative framework is inspired by 3DShape2VecSet (Zhang et al., 2023), a state-of-the-art two-stage approach for 3D shape representation and generation . In the first stage, a variational autoencoder (VAE) (Kingma and Welling, 2013) encodes semantic point clouds sampled from the shape surface, learning a distribution of robust latent semantic shape

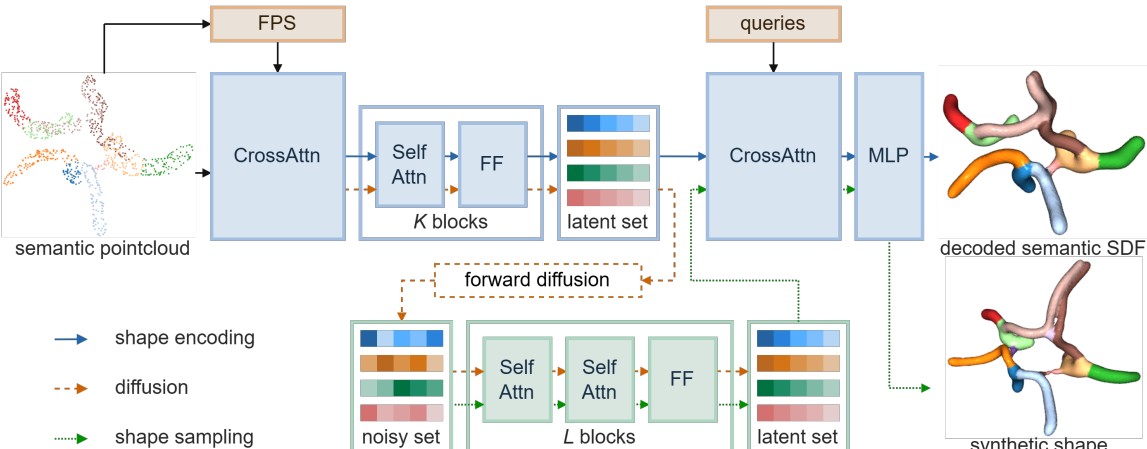

Figure 2: Schematic overview of the vessel generation framework: A semantic pointcloud is encoded to a shape representation using self-attention and feed-forward (FF) blocks. The shape representation is decoded and the SDF values are predicted for the query points. The diffusion model generates shape representations that are decoded to synthetic shapes.

representations that are then decoded into semantic SDFs. In the second stage, a latent diffusion model (Rombach et al., 2022) samples latent shape representations from the VAE's prior distribution. Latents sampled from this distribution are then decoded into a semantic SDF that represents a synthetic vascular tree.[1]

## 2. Method

Our generative framework comprises two models. Section 2.1 details the implementation of the variational autoencoder (VAE) for shape-to-semantic signed distance field (SDF) mapping. The VAE includes an encoder that learns robust latent shape representations for synthetic shape sampling and a decoder that outputs a conditional SDF corresponding to a given latent. Section 2.2 describes the shape latent diffusion model used to sample latents for synthetic shape generation. An overview of the framework is provided in Figure 2.

### 2.1. Semantic Shape Autoencoder

We parameterize a shape as a point cloud of $N$ points $\mathbf{p}_i$ that lie on the shape surface with corresponding one-hot encodings $\mathbf{h}_i$ of semantic labels. We use a VAE with a transformer architecture (Vaswani et al., 2017) to encode the pointcloud to a set of $M$ latent vectors $\mathbf{z}_i$, where $M < N$. Next, the latent-set is decoded to an SDF that represents the zero iso-surface of the encoded shape.

**Signed Distance Fields**   SDFs implicitly represent the surface of shapes as a functions $f(\mathbf{p}) = d$ that outputs the signed distance $d$ from a spatial coordinate $\mathbf{p}$ to the shape surface,

---

1. Our code is available here.

where $d$ is negative for points inside the shape volume. [2] The shape surface is defined by the zero level-set, i.e., all coordinates $\mathbf{p}_i$ where $f(\mathbf{p}_i) = 0$. SDFs satisfy the Eikonal equation, $||\nabla_{\mathbf{p}} f|| = 1$, and for coordinates on the surface, the gradient $\nabla_{\mathbf{p}} f$ corresponds to the surface normal vector $\mathbf{n^P}$. The Eikonal constraint acts as an inductive bias for implicitly regularizing SDF learning (Gropp et al., 2020). By leveraging this constraint, it becomes unnecessary to know $d$ for points off the surface, eliminating the need for ground-truth signed distances. As a result, SDFs can be learned in a self-supervised manner by enforcing $d = 0$ on surface points and ensuring that the gradients of both on-surface and off-surface points satisfy the Eikonal equation.

**Shape Encoding**   The input to the encoder is a set of $N$ vectors $\mathbf{x}$ that are the concatenation of points $\mathbf{p}$ and one-hot encoded semantic vessel labels $\mathbf{h}$. Following (Zhang et al., 2023), we use furthest-point sampling (FPS) to obtain a lower-resolution set of $M$ vectors $\mathbf{y}$ that are then used to gather downsampled feature vectors from the input using cross-attention:

$$\text{CrossAttn}(\mathbf{y}_i, \{\mathbf{x_1}, \cdots, \mathbf{x_N}\}) = \sum_j a_{ij} \mathbf{v}(\mathbf{x}_j) \quad \text{and} \quad a_{ij} = \text{softmax}\left(\frac{\mathbf{q}(\mathbf{y}_i)^T \mathbf{k}(\mathbf{x}_j)}{\sqrt{D}}\right), \quad (1)$$

where $\mathbf{q}, \mathbf{k}, \mathbf{v} \in \mathbb{R}^D$ are the query, key, and value functions used in the attention mechanism. [3] Note that Equation 1 becomes self-attention when $\mathbf{x} = \mathbf{y}$. The feature vectors are then mapped using a series of self-attention blocks followed by a linear map to a set of $C'$-dimensional $\boldsymbol{\mu}$ and $\log \boldsymbol{\sigma}^2$ from which the $M$ shape latents $\mathbf{z}$ are sampled.

**Shape Decoding**   The decoder maps latent representations to $C$-dimensional feature vectors, which are then interpolated by query coordinate points using cross-attention. Each interpolated coordinate is subsequently mapped to a signed distance and semantic label via a two-layer linear mapping with GELU activation. We evaluate the surface point distances $\tilde{d}$ and one-hot semantic labels $\tilde{\mathbf{h}}$ predicted by the decoder $g$ for query points surface points $\mathbf{p}$ and off-surface points $\mathbf{o}$ with the following objectives:

$$\mathcal{L}_{\text{Surface}} = |\tilde{d}|, \quad \mathcal{L}_{\text{Eikonal}} = (||\nabla_{\{\mathbf{o}, \mathbf{p}\}} g|| - 1)^2, \quad \text{and} \quad \mathcal{L}_{\text{Normal}} = \langle \nabla_{\mathbf{p}} g, \mathbf{n_p} \rangle, \quad (2)$$

and the predicted labels $\tilde{\mathbf{h}}$ as

$$\mathcal{L}_{\text{MSE}} = \text{MSE}(\mathbf{h}, \tilde{\mathbf{h}}). \quad (3)$$

Here, $|\cdot|$ denotes the $L1$-norm, $||\cdot||$ denotes the $L2$-norm, $\langle \cdot \rangle$ indicates cosine-similarity, and MSE is the mean squared error. We use MSE because the semantic SDF represents both signed distance and semantic class information as a unified vector field, making the task naturally suited for regression. Off-surface points $\mathbf{o}$ are generated by adding noise sampled from a Gaussian distribution with a standard deviation of 0.3 to the surface points. As the SDF defines the shape surface boundary, semantic labels are evaluated only for surface points, excluding a background label for off-surface points.

---

2. We follow the convention where $d$ is negative inside the shape volume to ensure that the surface normal vectors point outward.

3. In practice, we first embed the set of concatenated coordinate and feature vectors $\mathbf{x}$ with a linear embedding before downsampling.

**Training Objective**   The complete objective of the VAE minimizing the following loss

$$\mathcal{L}_{\text{VAE}} = \mathcal{L}_{\text{Surface}} + \lambda_1 \mathcal{L}_{\text{Eikonal}} + \lambda_2 \mathcal{L}_{\text{Normal}} + \lambda_3 \mathcal{L}_{\text{KL}} + \lambda_4 \mathcal{L}_{\text{MSE}}, \tag{4}$$

where $\mathcal{L}_{\text{KL}}$ is the KL-regularization of the latents $\mathbf{z}_i$ and the $\lambda_i$ weigh the contributions of the individual loss terms. The $\mathcal{L}_{\text{MSE}}$ term can be omitted if semantic labels are not available or not required.

### 2.2. Shape Latent Diffusion

Generating synthetic shapes by sampling latents $\mathbf{z}$ directly from the Gaussian prior often results in poor shape reconstructions because VAEs are prone to the prior-hole problem in large and complex latent spaces (Vahdat et al., 2021). This means that certain regions of the latent space do not hold any meaningful information. The prior-hole problem can be addressed by using an auxiliary sampling model $\phi$ that learns to sample latents exclusively from regions that yield high-quality reconstructions. To achieve this, we use latent-diffusion (Rombach et al., 2022) combined with a transformer architecture and optimize the mean-squared error between noisy and denoised latent sets:

$$\mathcal{L}_{\text{Denoise}} = \text{MSE}(\phi(\mathbf{z} + \epsilon, t), \mathbf{z}), \tag{5}$$

where $\epsilon \sim \mathcal{N}(0, t)$ is noise sampled at a given noise-level $t$. Details of the pre-conditioning diffusion methodology used in our approach can be found in (Karras et al., 2022).

## 3. Results and Evaluation

This section presents and evaluates the results of our semantic generative model. We detail our experimental setup in Section 3.1 and present and evaluate our results in Section 3.2.

### 3.1. Experimental Setup

We describe the datasets used for evaluation, the evaluation methodology and baseline models, as well as the training and sampling setup.

**Datasets**   We evaluate our model on two distinct vascular geometry datasets. TopCoW (Yang et al., 2024) contains 125 semantic segmentations of variations of the circle of Willis (CoW). An anatomical, semantic map of the CoW vasculature is provided in Appendix A. Next. VascuSynth (Hamarneh and Jassi, 2010) consists of 120 synthetic vascular trees and offers trees with a large variety in number of bifurcations per tree, but does not contain any semantic information. For both datasets, we sample 200,000 points from each shape surface and normalize them globally to a $[-1, 1]$ bounding box.

**Evaluation Methodology**   Generative models should produce samples that are representative of the real data, diverse, and unique—that is, not present in the training set. To assess representativeness, diversity, and uniqueness, we adopt a two-fold evaluation strategy. First, since our primary objective is to generate semantic cerebral vessel trees, we evaluate performance on the TopCoW dataset. We use the method from Kuipers et al. (2024) as our baseline, as it also generates semantic vessel trees. With access to semantic labels,

we assess representativeness and diversity by comparing the distributions of vessel length, average radius, and tortuosity between synthetic and real samples. Second, we evaluate on VascuSynth and compare against TrIND (Sinha and Hamarneh, 2024), a recent method that similarly uses implicit neural representations to generate vessel trees. As VascuSynth and TrIND lack semantic labels, we assess generative performance using global shape metrics: 1-nearest neighbor accuracy for representativeness and coverage for diversity. Details on these metrics are provided in Appendix E. Finally, we evaluate the ability of our model to generate unique vessel trees on both TopCoW and VascuSynth.

**Model Training and Sampling** We sample 2048 points from the shape surface as input to the VAE, which are subsequently downsampled to 256 points using cross-attention. 2048 additional surface points and 1024 off-surface points are sampled for calculating the loss. Both VAE and latent-diffusion models utilize six self-attention blocks. Detailed model architectures are provided in Appendix C. All models are trained with a batch size of 16. The VAE is trained for 9,000 epochs with a linear learning rate schedule, starting at $1 \times 10^{-6}$, increasing to $1.5 \times 10^{-4}$ over the first 200 epochs, and then decreasing to zero. The losses in Equation 4 are weighted for equal magnitude, with $\lambda_1 = \lambda_2 = 0.1$, $\lambda_3 = 1 \times 10^{-3}$, and $\lambda_4 = 1$. We apply random rotations of up to $\pm 0.1$ radians to the point clouds during VAE training, which significantly improves reconstruction quality. The latent-diffusion model is trained for 6,000 epochs on TopCoW and 12,000 epochs on VascuSynth. The same learning rate schedule is applied, with a maximum of $1 \times 10^{-4}$. All models require up to 2 hours and 4 GB of memory when trained on an NVIDIA TITAN Xp GPU. Synthetic trees are sampled in 100 steps with $\rho = 8$ and S_churn $= 25$. We refer to Karras et al. (2022) for more details on the sampling algorithm and its hyperparameters. Meshes are extracted from the zero level-set of the SDF using the marching cubes algorithm (Lorensen and Cline, 1998). To ensure topological consistency, i.e., the absence of disconnected vessel segments, we post-process the generated meshes to retain only the largest connected component, or the two largest in the case of TopCoW.

### 3.2. Generative Performance Analysis

**Semantic Vessel Tree Generation** We generate a set of semantic TopCoW vessel trees and compare our approach to Kuipers et al. (2024). Our qualitative analysis of the vessel tree quality in Figure 3 shows that our method successfully generates the circle of Willis anatomy. The different variations of the circle of Willis are well represented in the synthetic samples. In contrast, the method by Kuipers et al. (2024) fails to properly construct the tree topology, resulting in unrealistic circle of Willis trees. This is primarily due to the rule-based algorithm failing to reconstruct the tree topology in the presence of excessive noise and inconsistencies in the generated point clouds. Such failures were observed in approximately 90% of the generated samples, while no such issues occurred with the proposed method, which directly generates the entire tree as a single signed distance field.

To further assess the quality of our synthetic semantic trees, we automatically extract the length, average radius, and tortuosity of each vessel by skeletonizing the generated SDFs. The radius at each centerline point is computed as the shortest distance from that point to the vessel surface. We compare the distributions of these vessel characteristics between the real and synthetic TopCoW trees. The results in Figure 4 reveal distinct geometric

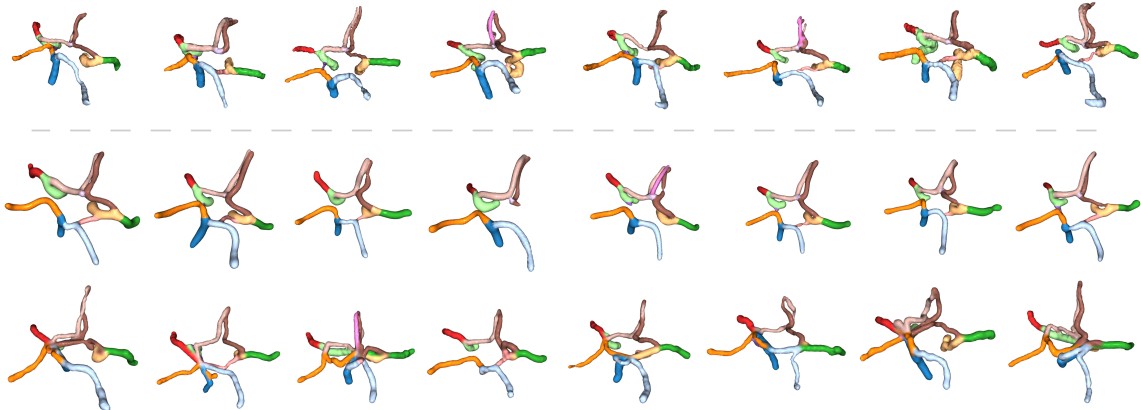

Figure 3: Real and synthetic semantic vessel trees from TopCoW. The **top** row displays real trees, the **middle** row shows synthetic trees generated by the proposed method, and the **bottom** row presents results from Kuipers et al. (2024), including failed tree topology reconstructions.

differences between the vessels in the real population. These differences are accurately reflected in the synthetic population, including outliers. This suggests that our model generates synthetic vessel trees that are diverse and representative of the real population. The L-Pcom, R-Pcom, and Acom seem to be the most challenging to generate, likely because these vessels only occur in a small subset of the trees in the real population. Additionally, the Acom in particular is short, and due to the lower voxel resolution used for synthetic shape sampling compared to the TopCoW segmentation resolution, the skeletonization often produces skeletons consisting of only one or two voxels, resulting in zero tortuosity. In Appendix B, we present vessel characteristics from (Kuipers et al., 2024), which reveal significant discrepancies from the real TopCoW trees, especially in length and tortuosity. These differences stem from the post-processing algorithm's failure to accurately reconstruct the tree topology.

**Baseline Tree Generation Performance** We compare our method to the results obtained by TrIND (Sinha and Hamarneh, 2024), an implicit neural shape (INS) method that generates non-semantic VascuSynth trees using occupancy grids as its implicit shape representation. We report 1-nearest-neighbor accuracy (1-NNA) and coverage (COV) to measure representativeness and diversity. The results in Table 1 show our model outperforming TrIND on both metrics. We attribute this to our use of a single encoder to encode all shapes, which enables weightsharing, resulting in a robust tree distribution that is more suitable for sampling compared TrIND's distribution of individually trained INS weights. Figure 5 demonstrates that our model can generate varied and high-quality tree structures. In Appendix 12, interpolation of the latent space reveals that our model learns a robust vessel tree representation.

**Synthetic Vessel Tree Uniqueness** We assess uniqueness by calculating similarity with the Chamfer distance for shapes within the train set (intra-distances) and between the synthetic and train sets (inter-distances). Figure 6 we observe a wide distribution of inter-

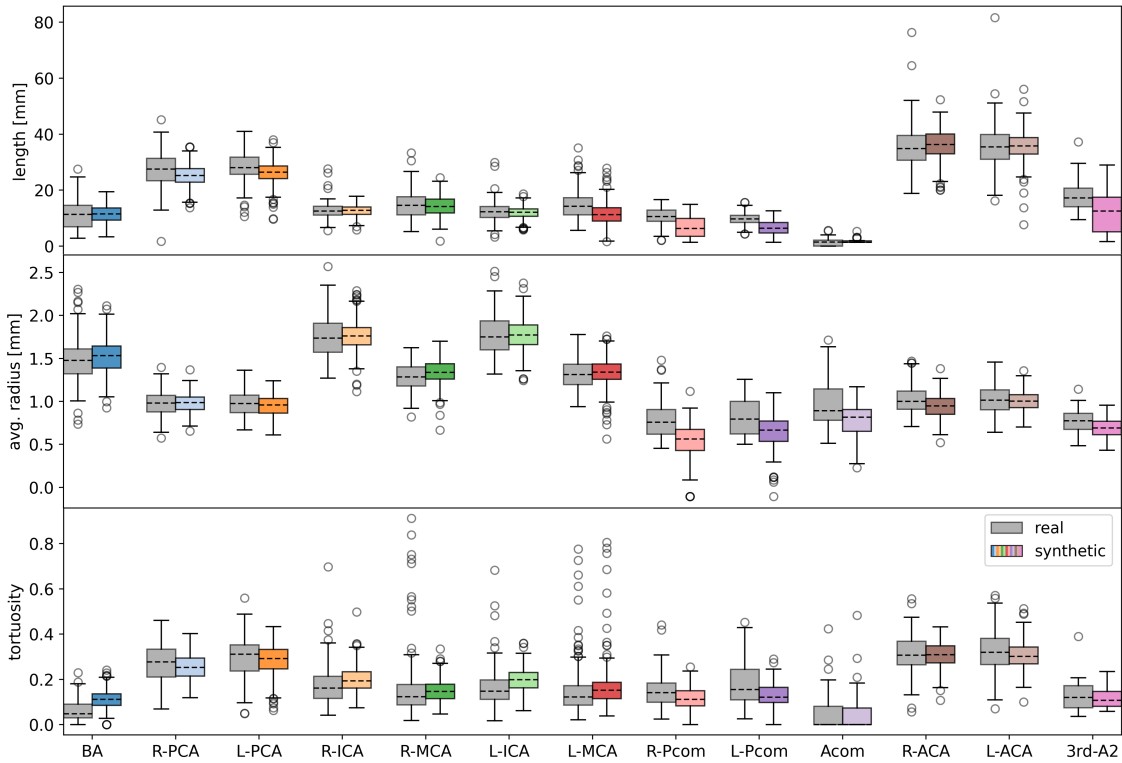

Figure 4: Comparison of the distributions of vessel length, average radius, and tortuosity for each individual vessel in the real and synthetic TopCoW trees.

Table 1: Quantitative comparison on VascuSynth. For 1-NNA, 50% is optimal. For COV, higher is better.

| metric | 1-NNA (%) | COV ↑ |
|---|---|---|
| TrIND | $87.4_{\pm 8.4}$ | $0.5_{\pm 0.1}$ |
| ours | $\mathbf{57.0}_{\pm 2.8}$ | $\mathbf{0.7}_{\pm 0.1}$ |

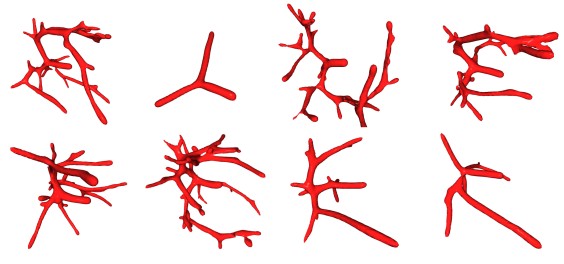

Figure 5: Synthetic VascuSynth trees generated with our model.

distances when compared to the intra-distances for VascuSynth, indicating a high degree of uniqueness. For TopCoW, the inter-distances are generally lower than the intra-distances. This indicates that the synthetic trees are less unique, likely due to the greater similarity among real TopCoW trees. As a result, the synthetic trees tend to be more "in-between" the real trees, leading to lower inter-distances. Nonetheless, Figure 7 demonstrates that our model is capable of generating unique trees for both TopCoW and VascuSynth.

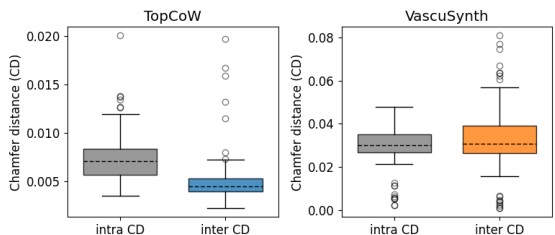

Figure 6: Intra and inter Chamfer distances (CDs) between most similar trees within the train set and between train and test sets.

Figure 7: Synthetic TopCoW and VascuSynth trees with most similar tree from the train set overlayed in gray.

## 4. Discussion and Conclusion

In this paper, we introduced a framework for representing and generating semantic vascular trees using signed distance fields in a fully self-supervised manner. Our results demonstrate that the proposed model can produce realistic synthetic vessel tree populations closely resembling real ones. However, quantitatively defining what constitutes a truly realistic vessel tree remains a significant challenge. While global shape metrics, such as 1-nearest-neighbor accuracy and coverage, provide some insights into a generative model's overall performance, they lack the precision needed to detect potential inaccuracies in individual vessels, given the intricate nature of vessel tree structures.

For our evaluation, we analyzed the characteristics of each individual synthetic vessel and compared them to the real vessels to assess the quality of the synthetic trees. However, the real trees used for comparison represent only a limited subset of all plausible vessel trees. In the context of small datasets, improving representativeness can lead to overfitting behavior, resulting in synthetic trees that are less unique and diverse. Thus, a trade-off exists between uniqueness, diversity, and representativeness. For the downstream task of stroke-treatment simulation, generating rare vascular structures with more challenging anatomy, such as higher tortuosity or smaller radii, could improve the robustness of novel device and treatment evaluations. Ultimately, the quality of the synthetic vessels should be determined based on how well they serve their intended downstream tasks.

Beyond quantitative analysis, qualitative evaluation by domain experts can offer valuable insights into the quality of the synthetic vessel trees. However, for in-silico clinical trials that require large virtual populations, manually assessing whether synthetic samples are suitable becomes unpractical. A promising future direction is to enable generative models to self-assess the quality of their outputs. Recent work by Islam et al. (2024) demonstrated that probabilistic signed distance fields can allow models to identify regions with potential artifacts through uncertainty awareness. In the context of shape generation, uncertainty awareness could lead to the automatic detection of inaccurate regions in the synthetic shapes.

In conclusion, our self-supervised method eliminates the need for post-processing algorithms to generate topologically complex and high-quality semantic cerebral vessel trees that are representative of real-world vessels, diverse, and unique.

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

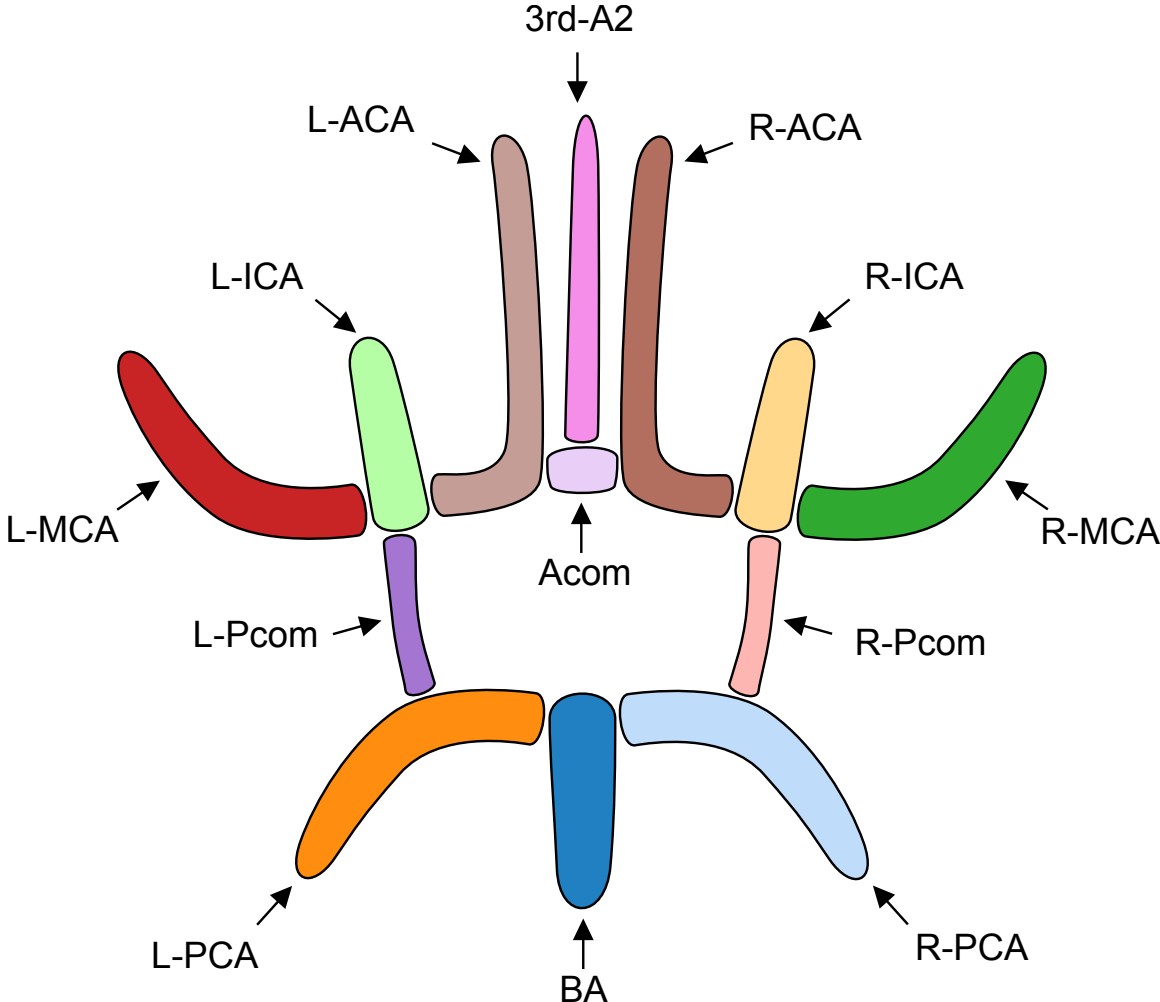

Figure 8: Anatomical map of the Circle of Willis as represented in TopCoW (Yang et al., 2024).

## Appendix A. TopCow-2024 Circle of Willis Map

Figure 8 presents a schematic overview of the complete Circle of Willis vascular structure from the TopCoW (Yang et al., 2024) dataset. The dataset includes many anatomical variations, often comprising only subsets of the vessels shown here. Most variability in TopCoW tree geometry arises from differing combinations of the left/right Pcom, Acom, and 3rd-A2 vessels.

## Appendix B. Vessel Characteristics from Kuipers et al. (2024)

Figure 9 presents the geometric vessel characteristics (length, average radius, and tortuosity) of synthetic TopCoW vessel trees generated by the method proposed in Kuipers et al. (2024). Consistent with the qualitative observations in Figure 3, substantial differences exist between synthetic and real trees, particularly in vessel lengths and tortuosities, which

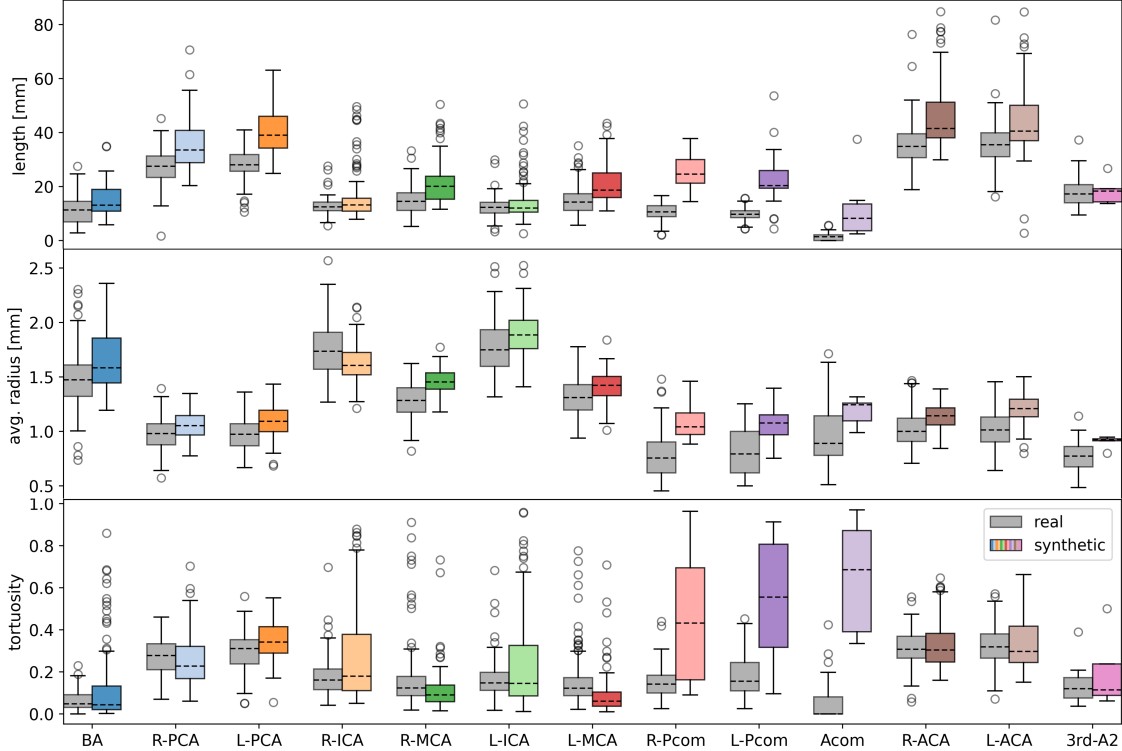

Figure 9: Comparison of the distributions of length, average radius, and tortuosity for individual vessels between real TopCoW trees and synthetic trees generated by the method of Kuipers et al. (2024).

are notably inaccurate in the synthetic data. The method by Kuipers et al. (2024) operates in two stages: first, generating the vessel tree outline as a semantic point cloud; second, applying a rule-based algorithm to reconstruct the tree topology from the unordered point cloud. This reconstruction involves sequencing points within individual vessels and then establishing bifurcation points between vessels. The algorithm's performance depends heavily on dense, equidistantly spaced points and well-defined vessel segments. Due to the complex topology of the Circle of Willis, this rule-based approach struggles to generalize, often requiring case-specific tuning to accurately reconstruct tree topology. Failures in reconstruction introduce sharp, anatomically improbable vessel angles, which account for the large discrepancies in length and tortuosity between synthetic and real trees.

## Appendix C. Model Architectures

**Variational Autoencoder Architecure** Figure 10 illustrates the architecture of our variational autoencoder. The encoder input is a semantic point cloud consisting of 3D $(x, y, z)$ coordinates and corresponding one-hot encoded semantic labels. This input is encoded into a set of shape latent variables. The decoder receives these shape latents along with query coordinate points and predicts both a signed distance and a semantic label for

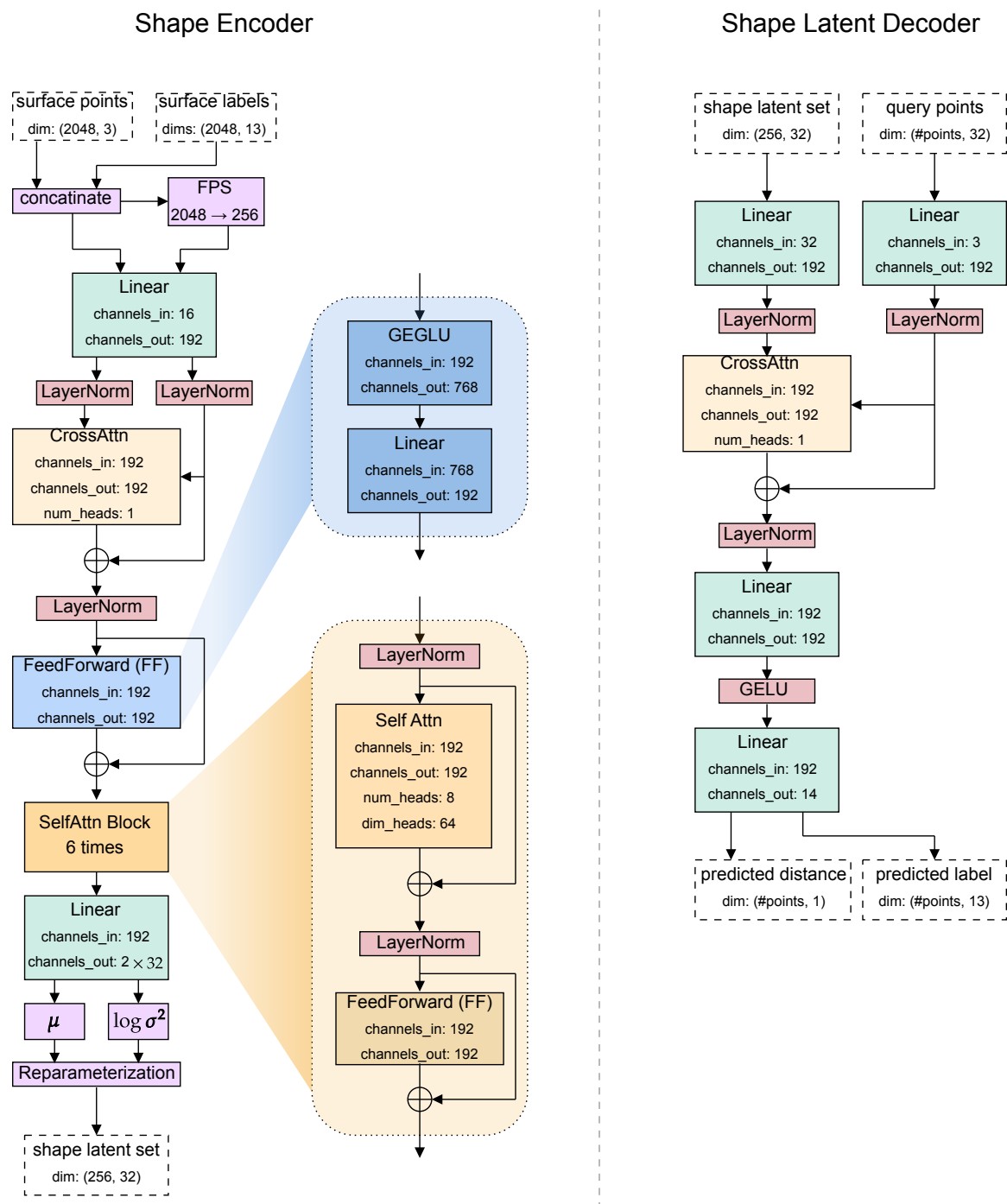

Figure 10: Architecture of the variational autoencoder.

each query. Thus, the decoder functions as a conditional semantic signed distance function, with the shape latents serving as the conditioning variables.

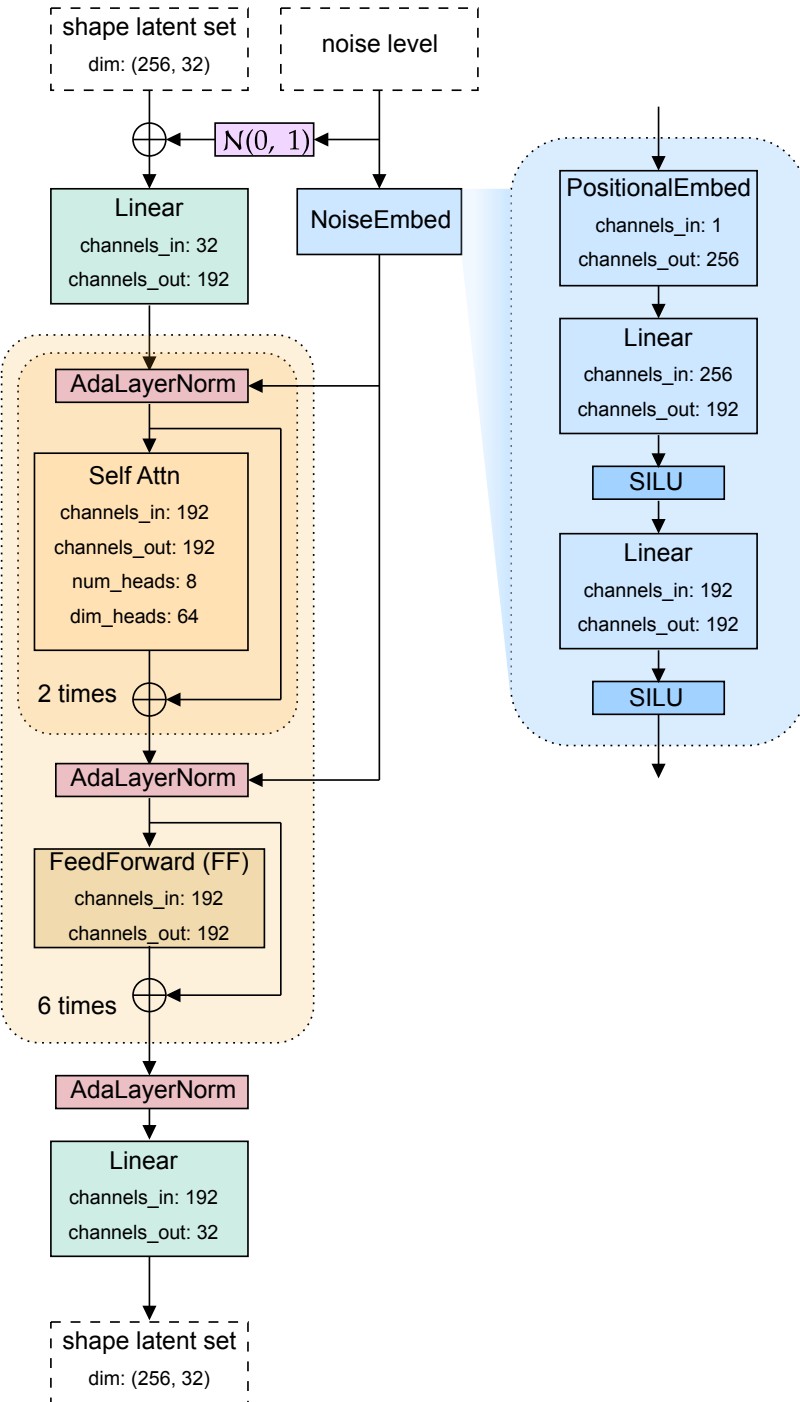

Figure 11: Architecture of the diffusion model.

**Shape Latent Diffusion Architecture**  Figure 11 illustrates the architecture of the shape latent diffusion model. The model takes a set of shape latents as input. During the forward diffusion process, a noise level is sampled and used to generate noise, which

is added to the shape latents. The model then denoises these noisy latents, conditioned on the sampled noise level. New shape latents can be generated by denoising noise drawn from a unit Gaussian. When decoded by the decoder network, these latents produce novel synthetic shapes.

## Appendix D. Interpreting the Latent-Diffusion Shape Space

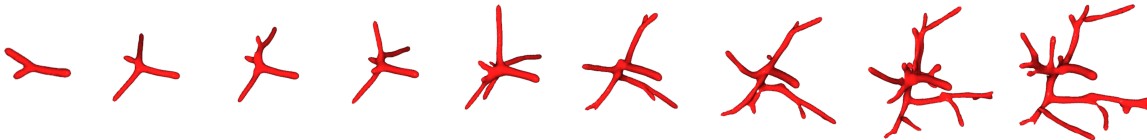

Figure 12: Interpolating the diffusion latent-space between a VascuSynth (Hamarneh and Jassi, 2010) tree with a single bifurcation and a tree with a large number of bifurcations.

To analyze the learned shape representations, we interpolate between two sampled VascuSynth (Hamarneh and Jassi, 2010) shapes: starting from a simple tree with a single bifurcation and ending with a complex tree featuring many bifurcations. We perform linear interpolation between the diffusion model's input noise vectors, denoising at each step. The resulting shapes over ten interpolation steps are shown in Figure 12. As we move through the latent space, the initial branches elongate and begin to bifurcate, indicating that the model has learned a robust representation of tree structures. The interpolation preserves the original shape while gradually increasing its complexity.

## Appendix E. 1-Nearest Neighbor Accuracy and Coverage

The 1-nearest neighbor accuracy (1-NNA) measures representativeness by quantifying how similar the real and synthetic shape distributions are. It classifies each shape based on the dataset of its nearest neighbor, using a distance metric—in our case, the Chamfer distance (CD) between point clouds. An accuracy of 50% indicates no distinction between real and synthetic distributions, as half of the real shapes are classified as synthetic and vice versa. Following Erkoç et al. (2023), we formulate 1-NNA between a set of reference shapes $S_r$ and a set of generated shapes $S_g$ as

$$1\text{-NNA}(S_r, S_g) = \frac{1}{|S_r| + |S_g|} \sum_{X \in S_r} \mathbf{I}[N_X \in S_r] + \sum_{Y \in S_g} \mathbf{I}[N_Y \in S_g], \qquad (6)$$

where $|\cdot|$ is set-cardinality, $\mathbb{I}$ is the indicator function, and $N_X$ the point cloud that is closest to $X$ from the union of the referene set and synthetic set:

$$N_X = \underset{K \in S_r \cup S_g}{\arg \min} \, \text{CD}(X, K). \qquad (7)$$

Coverage (COV) evaluates diversity by finding the nearest real neighbor for each synthetic sample and computing the ratio of unique real neighbors to the total number of synthetic

samples. We formulate COV between a set of reference shapes $S_r$ and a set of generated shapes $S_g$ as

$$\mathrm{COV}(S_r, S_g) = \frac{1}{|S_r|} \left| \left\{ \underset{X \in S_r}{\arg\min} \, \mathrm{CD}(X, Y) | Y \in S_g \right\} \right|. \tag{8}$$

