# OpenReview forum: "Self-Supervised Synthetic Cerebral Vessel Tree Generation using Semantic Signed Distance Fields"
_MIDL.io/2025/Conference — MIDL 2025 Oral_

### Official Review · Reviewer_EDwa · 2025-02-18

**Confidence:** 4
**Preliminary Rating:** 5
**Recommendation:** Oral

**Summary:**

This paper builds upon a previous work of the same author in last year's edition of the conference and presents a big improvement of their deep learning algorithm to vessel tree generation. They improved their method to make it work on complex tree topologies, such as in the circle of Willis.

**Strengths:**

The paper is well-written, the problem and its difficulties are clearly stated and the method and results are well-described. Such methods are valuable in several areas of research/clinical work where a precise algorithm to reconstruct vessels is important.

**Weaknesses:**

There are not many weaknesses in this work that I could find. Maybe one is that it only compares with the previous work of the author. Maybe it would be of interest to compare it with at least one other similar algorithm, if possible. It would also be of interest to provide a little more on the authors's previous work, for those not aware of it, and in particular what differentiate between the two. It is only briefly mentioned, but it would help to know a bit more about what makes the previous version fail on complex topologies and how this one work for example.

**Detailed Comments:**

I don't have much minor improvements for this paper, apart from what I wrote above.

**Justification Of The Preliminary Rating:**

It is a good paper, well presented that is worth publishing, as a major improvement of previous work at the same conference last year. I hope to see another one next year, maybe on not just on test data, but with real clinical application.

**Questions To Address In The Rebuttal:**

I would just like to see a little more context/comparison with previous works to put this one into perspective if possible, as mentionned above.

**Special Issue:**

No

---

> ### Author Response · Authors · 2025-03-07
>
> We thank the reviewer for their feedback and valuable suggestions.
>
> **WN1 & Q1**: There are not many weaknesses in this work that I could find. Maybe one is that it only compares with the previous work of the author. Maybe it would be of interest to compare it with at least one other similar algorithm, if possible. It would also be of interest to provide a little more on the authors's previous work, for those not aware of it, and in particular what differentiate between the two. It is only briefly mentioned, but it would help to know a bit more about what makes the previous version fail on complex topologies and how this one work for example.
>
> I would just like to see a little more context/comparison with previous works to put this one into perspective if possible, as mentionned above.
>
> **ANSWER**: We recognize that that additional context for previous works would improve the motivation for our evaluation methodology. We address this two fold.
>
> 1. We added more context on previous works in the introduction section and address more explicitely why previous works are not applicable to our problem setting of generating complex semantic vessel tree meshes without toplogical restrictions. Concretely, prior work is limited in tree topology by ,e.g., not supporting bifurcations, not supporting looping branches, and not supporting multiple disconnected trees – all properties observed in the circle of Willis (COW). Prior methods also do not provide any semantic information.  Two recent prior works come close and are used as our baseline comparisons. Sinha 2024 similarily generate meshes using implicit neural representations but do not generate semantic information. Kuipers 2024 generate centerlines with semantic information, but the tree-graph topology is constructed separately, which does not generalize well to more complex trees such as the COW.
>
> 2. We added more context in the results section to better motivate our experimental design and choices for baseline comparisons (Sinha 2024 and Kuipers 2024). Furthermore, we discuss in more detail the method from Kuipers 2024 to provide more perspective on where/why it fails on properly generating the COW and why the proposed method does not suffer from the same problem. In addition, the provide additional details in the Appendix where we add a section showing TopCoW vessel characteristics results from Kuipers 2024 and compare them to the real vessels.
>
> **CHANGES TO MANUSCRIPT:**
>
> To the introduction section 1, we revised the second paragraph to the following:
>
> *"Several methods for generating 3D vascular geometry have been proposed in recent literature. \citet{danu2019deep} utilize an image-based generative approach and represent the geometry as discrete 3D voxel occupancy grids, limiting the resolution of the generated vessels. \citet{wolterink2018blood} generate single-branch centerline graphs sequentially using a generative adversarial network (GAN), but do not support bifurcations. Expanding on this concept, \citet{feldman2023vesselvae} generate vessel centerline tree graphs using a recursive variational autoencoder that does supports bifurcations. However, the method is limited by its recursion depth and does not support looping structures or multiple disconnected graphs, structures that do occur in cerebral vascular structures such as the circle of Willis. As such, these methods are not directly applicable for generating these cerebral vessel trees. In contrast, \citet{sinha2024representing} represent 3D vascular geometry using implicit neural representations (INRs) with occupancy fields, making this method free of topological restrictions.*
>
> *Besides topological restrictions, a major limitation of the aforementioned methods is the lack of semantic labeling of the individual vessels in the generated trees, which is crucial in computational stroke treatment models for the automatic placement of thrombi in specific vessels and locations. Additionally, having access to semantic labels allows for more robust evaluation of the synthetic vessel trees by assessing the quality of each individual vessel within the tree. As a result, \citet{kuipers2024generating} introduced a pointcloud-based diffusion approach for generating semantic cerebral vessel trees. However, this method requires separate post-processing algorithms to reconstruct the vessel tree topology for which we show that it does not generalize well to topologically complex trees."*

---

> > ### Author Response · Authors · 2025-03-07
> >
> > **CHANGES IN MANUSCRIPT**
> >
> > In the results section 3.1 we added the following to motivate our experimental design:
> >
> > *“\paragraph{Evaluation Methodology} Generative models should produce samples that represent the real data, are diverse, and unique, i.e., not seen in the training data. To assess representativeness, diversity, and novel shape generation, our evaluation follows a two-fold approach. First, since our primary goal is to generate semantic cerebral vessel trees, we evaluate our model on TopCoW. We use the method from \citet{kuipers2024generating} as our baseline, since it also generates semantic vessel trees. With semantic vessel labels, we assess representativeness and diversity by comparing the distributions of length, average radius, and tortuosity of the synthetic data to real data. Next, we evaluate on VascuSynth and compare against TrIND \cite{sinha2024representing}, a recent method that similarly uses implicit neural shapes for representing and generating vessel trees. Since VascuSynth and TrIND do not utilize semantic labels, we evaluate generative performance with global shape metrics, where 1-nearest neighbor accuracy and coverage are used to measure representativeness and diversity, respectively. We refer to Appendix E for more details on these metrics. Finally, we assess our model's ability to generate unique shapes not observed in the training data on both TopCoW and VascuSynth.”*
> >
> > In the results section 3.2 we added the following context to the failure of Kuipers 2024:
> >
> > *“This is mainly the result of the rule-based algorithm failing to reconstructs the tree topology due to too much noise and inconsistency in the generated point clouds.} We observed these failures in approximately 90\% of the generated samples, \revision{whereas we did not observe such failures with the proposed method since it directly generates the entire tree as a single signed distance field.”*
> >
> > We added Appendix B which includes vessel characteristics results obtained from Kuipers 2024, showing the failure to properly capture the TopCoW vessel geometry:
> >
> > *“In Figure 9 we show the geometric vessel characteristics (length, average radius, and tortuosity) of the synthetic TopCoW vessel trees generated by the method proposed in \cite{kuipers2024generating}. As was observed from the qualitative results in Figure 3, we see large differences in geometric characteristics between the synthetic and real trees. In particular, the lengths and tortuosities of the synthetic vessels are highly inaccurate. The method by \cite{kuipers2024generating} consists of two stages. First, the vessel tree outline is generated as a semantic point cloud. Next, because the point cloud consists of a set of unordered points, a second rule-based algorithm is used that reconstructs the tree topology. This algorithm first sequences the points of the individual vessels within the tree, and then constructs the bifurcation points between these vessels. The rule-based algorithm thus heavily depends on dense point clouds with equidistantly spaced points and clearly distinct individual vessel segments. The complex topological nature of the circle of Willis thus prevents the rule-based algorithm from generalizing, requiring careful tuning on a case-by-case basis to properly reconstruct the tree topology. In the cases where reconstruction fails, sharp and improbable angles of the vessel geometry are introduced, resulting in the large discrepancy between the lengths and tortuosities of the synthetic trees and real trees.”*
> >
> > **OTHER:**
> >
> > “It is a good paper, well presented that is worth publishing, as a major improvement of previous work at the same conference last year. I hope to see another one next year, maybe on not just on test data, but with real clinical application.”
> >
> > We hope so as well!

---

### Official Review · Reviewer_8E2w · 2025-02-18

**Confidence:** 4
**Preliminary Rating:** 4
**Recommendation:** Oral
**Final Rating:** 5

**Summary:**

The authors show a novel framework for the generation of vessel trees, specifically of the circle of Willis with a very complicated shape. The framework integrates a semantic VAE and shape latent diffusion. The work is very exciting and, moreover, a follow-up part of the project presented at MIDL one year ago. Overall, the quality of research, text, and figures is very high, however, I have some questions about model assessment.

**Strengths:**

The paper is well-written and with a clear structure.
The proposed model is fully explained, figures and charts are helpful and easy to understand.
Authors showed the outperformance of model over other methods.

**Weaknesses:**

The assessment of the model performance is a bit messy, please see my comments below.
There are also some other minor comments and a code release will be helpful.

P.S. The system requires more characters, so I wrote this part - there are no more weaknesses from my side of view.

**Detailed Comments:**

How were 1024 off-shape points selected?

Can authors provide a Github repo with code?

Can authors speculate what new metric can be proposed for the detection of failed topology? I guess that all metrics from Figure 4 are within ranges of real cases even for a comparison study (Kuipers 2024), and only visual assessment can help in separating bad synthetic cases.

Do authors plan to use these synthetic meshes to simulate blood flow?

Can the proposed methods generate non-closed meshes as in Figure 1?

Typos – ‘compare the analyze the characteristics’, ‘2048 and 1024 surface and off-surface points’

‘A major limitation of these methods is the lack of semantic information in the generated trees, which is crucial, in computational stroke treatment models require for automatic placements of thrombi in specific vessels. Additionally, it allows for more robust evaluation of the synthetic vessel trees by assessing the quality of each separate vessel in the tree.’ – please correct these sentences, it’s not clear what the semantic information is, why it’s crucial, and it looks like ‘lack of semantic information’ allows more robust evaluation.

**Justification Of The Final Rating:**

I thank the authors for addressing all my questions. The paper's quality is great, with beautiful figures, and I recommend this article for the oral talk and a special call of journal (MELBA Special Issue).

**Justification Of The Preliminary Rating:**

I appreciate the authors for presenting a well-organized and relevant paper. It's always a pleasure to review such work, and I'm overall very satisfied with the content. However, a few minor edits and a review of the assessment section would further enhance it.

**Questions To Address In The Rebuttal:**

The assessment design is a bit messy, it should be stated somewhere that only one dataset has labels, and, therefore, the relevant metrics (radius, etc) can be calculated for each type of vessel only for this dataset.

Also, 1-NNa and COV are calculated only for the second dataset and were compared with a different baseline, not Kuipers 2024 (Table 1). Can authors show these metrics for Kuipers 2024? Or state that they are similar but Kuipers 2024 method requires a manual check?
What about the Chamfer distance for baseline methods (Kuipers 2024, TrIND)?

To summarize – the authors showed some anatomy-based metrics for the labelled dataset, which is fine and clear, but then proposed a range of assessments based on different metrics (1-NNA, COV, Chamfer dist.) for two different baselines (Kuipers 2024 and TrIND) and two different datasets. Overall, it should be 3 metrics by 3 methods (proposed and two baselines) by 2 datasets = 18 values, but authors showed only part of them without explanation of such a selection. Maybe, it will be helpful to add a separate paragraph about assessment metrics in Methods.

‘1-nearest-neighbor accuracy (1-NNA) and coverage (COV)’ – what do these metrics show? 50% optimal value should be somewhere in the text.

What does S_churn stand for?

‘To further assess the quality of our synthetic semantic trees, we extract the length, average radius, and tortuosity of each individual vessel by skeletonizing the generated SDFs.’ – is it an automatic or manual process? How was the radius assessed from the skeleton?

**Special Issue:**

Yes

---

> ### Author Response · Authors · 2025-03-07
>
> We thank the reviewer’s detailed comments and valuable feedback.
>
> **C1**: How were 1024 off-shape points selected?
>
> ANSWER: The off-shape points are selected by first sampling 1024 points from the shape-surface and randomly sampling gaussian noise with a zero mean and 0.3 standard deviation that is then added to the surface points. This results in more points closer to the surface, where the quality of the predicted SDF is the more important.
>
> **C2**: Can authors provide a Github repo with code?
>
> **ANSWER**: We will provide a Github repo with our code.
>
> **CHANGES TO MANUSCRIPT:**
>
> We added a footnote in the introduction with a link to our code:
>
> *"Our code is available here."*
>
> **C3**: Can authors speculate what new metric can be proposed for the detection of failed topology? I guess that all metrics from Figure 4 are within ranges of real cases even for a comparison study (Kuipers 2024), and only visual assessment can help in separating bad synthetic cases.
>
> **ANSWER:** A possible metric would be some form of negative log-likelihood evaluation of the final output tree (so after reconstruction). More specifically, we could consider the reconstructed tree as a new point cloud and measure the likelihood of that point cloud originating from the distribution of ground truth point clouds learned by the generative model. Because qualitative results show that the geometry of the reconstructed trees is highly implausible, such a likelihood measure would reflect this, allowing for the detection of failed samples. Moreover, we now added TopCoW vessel characteristic distributions obtained from Kuipers 2024 (see **Q2-Q3**) that do show a large discrepancy between the synthetic tree characteristics compared to those observed in the real data.
>
> **C4**: Do authors plan to use these synthetic meshes to simulate blood flow?
>
> **ANSWER**: Yes. We plan to use the synthetic meshes to simulate mechanical thrombectomy and aspiration, and blood flow or pressure can also be part of these simulations, e.g., for pushing the virtual clot through the vessels.
>
> **C5**: Can the proposed methods generate non-closed meshes as in Figure 1?
>
> **ANSWER**: No, currently the proposed methods only generate fully-closed meshes. This means that meshes that are open will be closed with additional geometry. This is because the signed distance field training objective is to learn a smooth closed surface. One solution if the generation of non-closed meshes is desired could be that, since our method allows for generating semantic information, vertices on the open ends such as in Figure 1 could be labeled accordingly. Then when generating synthetic meshes, the vertices belonging to the regions where the model closed are simirily labeled and can be removed algorithmically. This would not require manual labour for post-processing the generated meshes if non-closed surfaces are desired.
>
> **CHANGES TO MANUSCRIPT:**
>
> We added the following to clarify this to the introduction:
>
> *“..., and will always yield a mesh that is fully closed.”*
>
> **C6**: Typos - ‘compare the analyze the characteristics’, ‘2048 and 1024 surface and off-surface points’.
>
> **ANSWER**: Thank you for spotting these typos, we fixed them in the revised manuscript.
>
> **CHANGES TO MANUSCRIPT:**
>
> Changed:
>
> *“2048 and 1024 surface and off-surface points”*
>
> To:
>
> *“2048 surface and 1024 off-surface points”*
>
> And changed:
>
> *"compare the analyze the characteristics "*
>
> To:
>
> *"analyze the characteristics "*

---

> > ### Author Response · Authors · 2025-03-07
> >
> > **C7**: A major limitation of these methods is the lack of semantic information in the generated trees, which is crucial, in computational stroke treatment models require for automatic placements of thrombi in specific vessels. Additionally, it allows for more robust evaluation of the synthetic vessel trees by assessing the quality of each separate vessel in the tree.’ – please correct these sentences, it’s not clear what the semantic information is, why it’s crucial, and it looks like ‘lack of semantic information’ allows more robust evaluation.
> >
> > **ANSWER**: In line with the other reviewers’ comments, we revised the paragraph to provide more context on the previous work and why this work is not applicable to our problem setting, which includes defining the semantic information. In our case, the semantic information are the labels of the individual vessels that make up the complete vessel tree. Having access to these labels allows for more robust evaluation of the synthetic vessel quality because it enables evaluation of individual vessel segments.
> >
> > **CHANGES TO MANUSCRIPT:**
> >
> > Changed:
> >
> > *"A major limitation of these methods is the lack of semantic information in the generated trees, which is crucial, in computational stroke treatment models require for automatic placements of thrombi in specific vessels. Additionally, it allows for more robust evaluation of the synthetic vessel trees by assessing the quality of each separate vessel in the tree. "*
> >
> > To:
> >
> > *“Besides topological restrictions, a major limitation of the aforementioned methods is the lack of semantic labeling of the individual vessels in the generated trees, which is crucial in computational stroke treatment models for the automatic placement of thrombi in specific vessels and locations. Additionally, having access to semantic labels allows for more robust evaluation of the synthetic vessel trees by assessing the quality of each individual vessel within the tree.”*
> >
> > **Q1**: The assessment design is a bit messy, it should be stated somewhere that only one dataset has labels, and, therefore, the relevant metrics (radius, etc) can be calculated for each type of vessel only for this dataset.
> >
> > **ANSWER**: We recognize that currently it is not entirely clear from the manuscript why the assessments for the two datasets differ. We address this by adding an introductory evaluation methodology section to the results section that motivates the differences in experimental design for the two datasets and corresponding baselines Sinha 2024 (TrIND) and Kuipers 2024.
> >
> > **CHANGES TO MANUSCRIPT:**
> >
> > We added the following paragraph to Section 3.1:
> >
> > *"\paragraph{Evaluation Methodology} Generative models should produce samples that represent the real data, are diverse, and unique, i.e., not seen in the training data. To assess representativeness, diversity, and novel shape generation, our evaluation follows a two-fold approach. First, since our primary goal is to generate semantic cerebral vessel trees, we evaluate our model on TopCoW. We use the method from \citet{kuipers2024generating} as our baseline, since it also generates semantic vessel trees. With semantic vessel labels, we assess representativeness and diversity by comparing the distributions of length, average radius, and tortuosity of the synthetic data to real data. Next, we evaluate on VascuSynth and compare against TrIND \cite{sinha2024representing}, a recent method that similarly uses implicit neural shapes for representing and generating vessel trees. Since VascuSynth and TrIND do not utilize semantic labels, we evaluate generative performance with global shape metrics, where 1-nearest neighbor accuracy and coverage are used to measure representativeness and diversity, respectively. We refer to Appendix E for more details on these metrics. Finally, we assess our model's ability to generate unique shapes not observed in the training data on both TopCoW and VascuSynth."*

---

> > ### Author Response · Authors · 2025-03-07
> >
> > **Q2-Q3:**
> >
> > Also, 1-NNa and COV are calculated only for the second dataset and were compared with a different baseline, not Kuipers 2024 (Table 1). Can authors show these metrics for Kuipers 2024? Or state that they are similar but Kuipers 2024 method requires a manual check? What about the Chamfer distance for baseline methods (Kuipers 2024, TrIND)?
> >
> > To summarize – the authors showed some anatomy-based metrics for the labelled dataset, which is fine and clear, but then proposed a range of assessments based on different metrics (1-NNA, COV, Chamfer dist.) for two different baselines (Kuipers 2024 and TrIND) and two different datasets. Overall, it should be 3 metrics by 3 methods (proposed and two baselines) by 2 datasets = 18 values, but authors showed only part of them without explanation of such a selection. Maybe, it will be helpful to add a separate paragraph about assessment metrics in Methods.
> >
> > **ANSWER: As per Q1, we address the motivation for our experimental design between the two datasets and baseline models in the results section.**
> >
> > Generating the semantic cerebral vessel trees is our main focus. Here, we compare to Kuipers 2024, as this method was likewise designed to generate semantic vessel trees. Because we have access to the semantic labels of each individual vessel, we can evaluate the quality of each individual vessels by their geometric characteristics (length, tortuosity, and average radius). Because the qualitative analysis shows that Kuipers 2024 failes to generate plausible trees at all. As a result, including global shape metrics such as 1-NNA and COV would not add much value because they less precise in assessing shape quality compared to measuring the geometric characteristics of each invidivual vessel within the trees.
> >
> > We evaluate on VascuSynth because it offers a challenging tree generation task. We do not use the Kuipers 2024 baseline because it requires semantic information which is not present in VascuSynth. Instead, we compare to Sinha 2024 which was built for generating VascuSynth tree because it is similar to our method as it also used implicit neural representations. Because we do not have access to semantic information, we follow the evaluation strategy from Sinha 2024 and report global 1-NNA and COV metrics. Even though these metrics are not that precise in measuring shape quality, they do offer some insights in overall generative quality between different methods.
> >
> > **CHANGES TO MANUSCRIPT:**
> >
> > See changes of **Q1** regarding the added introduction to Section 3 on experimental design.
> >
> > Additionally, we added the quantitative results comparing Kuipers 2024 vessel characteristics to those of the real vessels on TopCoW to Appendix B with the following text:
> >
> > *“In Figure 9 we show the geometric vessel characteristics (length, average radius, and tortuosity) of the synthetic TopCoW vessel trees generated by the method proposed in \cite{kuipers2024generating}. As was observed from the qualitative results in Figure 3, we see large differences in geometric characteristics between the synthetic and real trees. In particular, the lengths and tortuosities of the synthetic vessels are highly inaccurate. The method by \cite{kuipers2024generating} consists of two stages. First, the vessel tree outline is generated as a semantic point cloud. Next, because the point cloud consists of a set of unordered points, a second rule-based algorithm is used that reconstructs the tree topology. This algorithm first sequences the points of the individual vessels within the tree, and then constructs the bifurcation points between these vessels. The rule-based algorithm thus heavily depends on dense point clouds with equidistantly spaced points and clearly distinct individual vessel segments. The complex topological nature of the circle of Willis thus prevents the rule-based algorithm from generalizing, requiring careful tuning on a case-by-case basis to properly reconstruct the tree topology. In the cases where reconstruction fails, sharp and improbable angles of the vessel geometry are introduced, resulting in the large discrepancy between the lengths and tortuosities of the synthetic trees and real trees.”*

---

> > ### Author Response · Authors · 2025-03-07
> >
> > **Q4**: ‘1-nearest-neighbor accuracy (1-NNA) and coverage (COV)’ – what do these metrics show? 50% optimal value should be somewhere in the text.
> >
> > **ANSWER and CHANGES TO MANUSCRIPT**:
> >
> > We added Appendix section E that explain the 1-NNA and COV metrics, including their formal definitions:
> >
> > *" The 1-nearest neighbor accuracy (1-NNA) quantifies representativeness by measuring the similarity between real and synthetic shape distributions. It classifies each shape based on the set its nearest neighbor belongs to according to a distance metrics. In our case, we use Chamfer distance (CD) as a distance metric between point clouds. A 50% accuracy indicates no distinction between real and synthetic distributions, as half of the real and synthetic shapes are classified as synthetic and real, respectively. Following \cite{erkocc2023hyperdiffusion}, we formulate 1-NNA between a set of reference shapes $S_r$ and a set of generated shapes $S_g$ as*
> >
> > \begin{align} \text{1-NNA}(S_r, S_g) = \frac{1}{|S_r| + |S_g|}\sum_{X \in S_r}\mathbf{I}[N_X \in S_r] + \sum_{Y \in S_g}\mathbf{I}[N_Y \in S_g], \end{align}
> >
> > *where $|\cdot|$ denotes set-cardinality and $N_X$ the point cloud that is closest to $X$ from the union of the referene set and synthetic set:*
> >
> > \begin{align} N_X = \argmin_{K \in S_r \cup S_g}\text{CD}(X, K). \end{align}
> >
> > *Coverage (COV) assesses diversity by identifying the nearest real neighbor for each synthetic sample and calculating the ratio of unique real neighbors to total synthetic samples. We formulate COV between a set of reference shapes $S_r$ and a set of generated shapes $S_g$ as*
> >
> > \begin{align} \text{COV}(S_r, S_g) = \frac{1}\{|S_r|}|{ \argmin_{X \in S_r}CD(XY)|Y \in S_g\}|. \end{align}"
> >
> > **Q5**: What does S_churn stand for?
> >
> > **ANSWER**: S_churn is a hyperparameter from the EDM diffusion framework we used from Karras 2022. This hyperparameter controls the stochasticity of the denoising process during inference. We now refer to the EDM literature for more details on these hyperparameters.
> >
> > **CHANGES TO MANUSCRIPT:**
> >
> > We added the following to section 3.1:
> >
> > *“We refer to \citet{karras2022elucidating} for more details on the sampling algorithm and its hyperparameters.”*
> >
> >
> >
> > **Q6**: ‘To further assess the quality of our synthetic semantic trees, we extract the length, average radius, and tortuosity of each individual vessel by skeletonizing the generated SDFs.’ – is it an automatic or manual process? How was the radius assessed from the skeleton?
> >
> > **ANSWER**: This is a fully automatic process. Once the centerline has been obtained, the radius at each centerline point is defined as the shortest distance from that point to the vessel surface. We will clarify that this is an automated process and how the radius is obtained in the revised manuscript.
> >
> > **CHANGES TO MANUSCRIPT:**
> >
> > We changed:
> >
> > *" we extract the length, average radius, and tortuosity of each individual vessel by skeletonizing the generated SDFs. "*
> >
> > To:
> >
> > *" we automatically extract the length, average radius, and tortuosity of each individual vessel by skeletonizing the generated SDFs, where the radius of each centerline point is calculated as the shortest distance of that point to the vessel surface "*

---

### Official Review · Reviewer_x6d2 · 2025-02-20

**Confidence:** 3
**Preliminary Rating:** 4
**Recommendation:** Poster
**Final Rating:** 5

**Summary:**

This work introduce a novel method for generating synthetic cerebral vessel tree. One of the main current issue in cerebral vascular network analysis is the lack of (annotated) data. Thus, this work appears very relevant in the current context.

The presented method use a VAE to encode a point cloud representation of the vessel tree and reconstruct a semantic signed distance field representing the vessel tree. Additionally, instead of sampling directly for this latent space the authors use a latent-diffusion model to generate latent vectors which lies in a "robust" region of the latent space. By doing so, they avoid the prior-hole problem inherent of VAEs.

In comparison, of other approaches in the SOTA this approach is end-to-end and don't need further post-processing.

The authors evaluate their method on a real world dataset and a synthetic dataset.

**Strengths:**

The proposed method is very relevant in regard to the current limitations in cerebral vascular network analysis. Moreover, the authors clearly motivate their work by a clinical challenge.

Furthermore, the technical novelty of the paper is significant and relevant. In particular, I really appreciated the use of the Eikonal equation to learn the SDF.

**Weaknesses:**

The main weakness of the paper is the evaluation part. I have the feeling that the evaluation lack comparison with other generation methods. Also in the context of vascular tree generation, it's important to assess the topology and connectivity of the generated trees.

**Detailed Comments:**

Overall, I really enjoyed reading this paper, the method is well motivated and context clearly introduced. Also, the method is well explained and easy for the reader to understand.

The field of vessels trees generation is still young, and this paper proposes a novel approach to the domain which is very valuable.

I think this paper is a fair contribution to MIDL and the community. However, I think the evaluation part should be strengthened before acceptation.

**Justification Of The Final Rating:**

I would like to thank the authors for their very interesting work and their commitment to address the reviewers comments.

The authors answers about their choices concerning the evaluation part were convincing. The novelty of their work and their setting make it difficult ton compare with existing methods that are designed for others settings. I have the feeling that the precisions added in the manuscript concerning this part and related work will make the contribution clearer for the readers.

Overall this is a very interesting work and well written paper that is definitely worth publishing at MIDL. Finally, I choose to rate this paper as a "strong accept".

**Justification Of The Preliminary Rating:**

The article present a novel and interesting method for generating vessel trees. Also, the article is well written, the motivations clearly introduced and the method well explained. The main limitation is the evaluation, which lack comparisons and experiments.

For these reasons, I choose to rate this paper as a weak accept. However, to me the evaluation should be improved before acceptance.

**Questions To Address In The Rebuttal:**

### Major Remarks

1 - I think It would be beneficial if the authors conduct a topological analysis on the generated vessel trees. For example, by computing the Betti numbers for both real vessel trees and generated vessel trees.

2 - Also, I think the evaluation lack some comparison. First, the comparison with the method from Kuipers et al. (2024) is only visual. The quantitive comparison (Fig. 4) should also include the work from Kuipers et al. (2024).

3 - Moreover, I think the authors should include other methods for vessel generation in the comparison. For example, maybe these articles can be interesting "3D Vessel Graph Generation Using Denoising Diffusion" Prabhakar et al. MICCAI 2024 and "Vesselvae: Recursive variational autoencoders for 3d blood vessel synthesis" Feldman et al. MICCAI 2023.

4 - I didn't understand why the authors use a $MSE$ loss to learn the semantic label $\textbf{h}$ and not a cross-entropy loss as it is a classification task.

### Minor Remarks

1 - In equation, (2) It would be clearer to define $\textbf{p}$ and $g$.

2 - If I understood correctly, to learn the VAE, points are sample in the surface and off-surface points are generated close to the surface points. It is sufficient to sample off-surface point only near the surface points to correctly learn the SDF ?

3 - In Figure 3, some real examples should be included to be compared with the generated vessel trees.

4 - The 1-NNA and COV metrics should be defined.

**Special Issue:**

No

---

> ### Author Response · Authors · 2025-03-07
>
> We thank the reviewer for their detailed comments and feedback.
>
> **Major Remarks**
>
> **Q1**: I think It would be beneficial if the authors conduct a topological analysis on the generated vessel trees. For example, by computing the Betti numbers for both real vessel trees and generated vessel trees.
>
> **ANSWER**: We post-process all generated meshes to only contain the largest connected component on VascuSynth and  the two largest components for TopCoW, as the latter contains such trees. This ensures that all generated trees are topologically consistent w.r.t. the zero-th Betti number and contain no disconnected vessel segments. All evaluation on the synthetic trees is performed on these post-processed meshes. We now more explicitely clarify this post-processing step in the revised manuscript.
>
> **CHANGES TO MANUSCRIPT:**
>
> To section 3.1 on model training and sampling, we added the following:
>
> *" To ensure that the generated vessel trees are topologically consistent, i.e., they do not contain any disconnected vessel segments, the generated meshes are post-processed to only consist of the largest connected component, or the two largest for TopCoW. "*
>
>
>
> **Q2**: Also, I think the evaluation lack some comparison. First, the comparison with the method from Kuipers et al. (2024) is only visual. The quantitive comparison (Fig. 4) should also include the work from Kuipers et al. (2024).
>
> **ANSWER**: Since the qualitative results show that the trees generated by Kuipers 2024 are plainly unusable and to keep figure 4 clear, we now include the additional results where we compare the vessel characteristics generated by Kuipers 2024 in the Appendix.
>
> **CHANGES TO MANUSCRIPT:**
>
> We added Appendix section B: Vessel Characteristics from Kuipers et al., 2024) with the following text:
>
> “In Figure 9 we show the geometric vessel characteristics (length, average radius, and tortuosity) of the synthetic TopCoW vessel trees generated by the method proposed in \cite{kuipers2024generating}. As was observed from the qualitative results in Figure 3, we see large differences in geometric characteristics between the synthetic and real trees. In particular, the lengths and tortuosities of the synthetic vessels are highly inaccurate. The method by \cite{kuipers2024generating} consists of two stages. First, the vessel tree outline is generated as a semantic point cloud. Next, because the point cloud consists of a set of unordered points, a second rule-based algorithm is used that reconstructs the tree topology. This algorithm first sequences the points of the individual vessels within the tree, and then constructs the bifurcation points between these vessels. The rule-based algorithm thus heavily depends on dense point clouds with equidistantly spaced points and clearly distinct individual vessel segments. The complex topological nature of the circle of Willis thus prevents the rule-based algorithm from generalizing, requiring careful tuning on a case-by-case basis to properly reconstruct the tree topology. In the cases where reconstruction fails, sharp and improbable angles of the vessel geometry are introduced, resulting in the large discrepancy between the lengths and tortuosities of the synthetic trees and real trees.”

---

> > ### Author Response · Authors · 2025-03-07
> >
> > **Q3**: Moreover, I think the authors should include other methods for vessel generation in the comparison. For example, maybe these articles can be interesting "3D Vessel Graph Generation Using Denoising Diffusion" Prabhakar et al. MICCAI 2024 and "Vesselvae: Recursive variational autoencoders for 3d blood vessel synthesis" Feldman et al. MICCAI 2023.
> >
> > **ANSWER:** We recognize that that additional context for previous works would improve the motivation for our evaluation methodology and why we did not include baselines beyond Sinha 2024 and Kuipers 2024. We address this two fold.
> >
> > 1. We now provide more context on previous works in the introduction section and address more explicitely why previous works are not applicable to our problem setting of generating complex semantic vessel tree meshes without toplogical restrictions. Concretely, the method by Feldman 2023 is limited in tree topology by, e.g., not supporting bifurcations, not supporting looping branches, and not supporting multiple disconnected trees – all properties observed in the circle of Willis (COW). This method also does not generate any semantic information. The method by Prabhakar 2024 does not generate vessel meshes. Instead, Prabhakar 2024 generates vessel tree graph topology.
> >
> > 2. We now provide more context in the results section to better motivate our experimental design and choices for baseline comparisons (Sinha 2024 and Kuipers 2024).
> >
> > **CHANGES TO MANUSCRIPT:**
> >
> > To the introduction section 1, we revised the second paragraph to the following:
> >
> > *" Several methods for generating 3D vascular geometry have been proposed in recent literature. \citet{danu2019deep} utilize an image-based generative approach and represent the geometry as discrete 3D voxel occupancy grids, limiting the resolution of the generated vessels. \citet{wolterink2018blood} generate single-branch centerline graphs sequentially using a generative adversarial network (GAN), but do not support bifurcations. Expanding on this concept, \citet{feldman2023vesselvae} generate vessel centerline tree graphs using a recursive variational autoencoder that does supports bifurcations. However, the method is limited by its recursion depth and does not support looping structures or multiple disconnected graphs, structures that do occur in cerebral vascular structures such as the circle of Willis. As such, these methods are not directly applicable for generating these cerebral vessel trees. In contrast, \citet{sinha2024representing} represent 3D vascular geometry using implicit neural representations (INRs) with occupancy fields, making this method free of topological restrictions.*
> >
> > *Besides topological restrictions, a major limitation of the aforementioned methods is the lack of semantic labeling of the individual vessels in the generated trees, which is crucial in computational stroke treatment models for the automatic placement of thrombi in specific vessels and locations. Additionally, having access to semantic labels allows for more robust evaluation of the synthetic vessel trees by assessing the quality of each individual vessel within the tree. As a result, \citet{kuipers2024generating} introduced a pointcloud-based diffusion approach for generating semantic cerebral vessel trees. However, this method requires separate post-processing algorithms to reconstruct the vessel tree topology for which we show that it does not generalize well to topologically complex trees. "*
> >
> > In the results section 3.1 we added the following to motivate our experimental design:
> >
> > *“\paragraph{Evaluation Methodology} Generative models should produce samples that represent the real data, are diverse, and unique, i.e., not seen in the training data. To assess representativeness, diversity, and novel shape generation, our evaluation follows a two-fold approach. First, since our primary goal is to generate semantic cerebral vessel trees, we evaluate our model on TopCoW. We use the method from \citet{kuipers2024generating} as our baseline, since it also generates semantic vessel trees. With semantic vessel labels, we assess representativeness and diversity by comparing the distributions of length, average radius, and tortuosity of the synthetic data to real data. Next, we evaluate on VascuSynth and compare against TrIND \cite{sinha2024representing}, a recent method that similarly uses implicit neural shapes for representing and generating vessel trees. Since VascuSynth and TrIND do not utilize semantic labels, we evaluate generative performance with global shape metrics, where 1-nearest neighbor accuracy and coverage are used to measure representativeness and diversity, respectively. We refer to Appendix E for more details on these metrics. Finally, we assess our model's ability to generate unique shapes not observed in the training data on both TopCoW and VascuSynth.”*

---

> > ### Author Response · Authors · 2025-03-07
> >
> > **Q4**: I didn't understand why the authors use a MSE loss to learn the semantic label  and not a cross-entropy loss as it is a classification task.
> >
> > **ANSWER:** Predicting the SDF is a regression task. We extend the SDF function to also represent additional fields, where each additional field represents one of the semantic labels. In doing so, the semantic label prediction remains a prediction of a scalar fields, i.e., the semantic SDF represents a vector field, for which a regression task remains appropriate. Treating the semantic label prediction as a classification task would likely work as well, but might be more appropriate when using occupancy prediction as the implicit shape representation, since that would involve classifying points as occupying the target shape or not.
> >
> > **CHANGES TO MANUSCRIPT:**
> >
> > We now provide additional context on the use of MSE by adding the following to section 2.1:
> >
> > *“We use MSE since the semantic SDF encodes signed distances and semantic information as a single vector field, optimizing it as a regression task.” *
> >
> > **Minor Remarks**
> >
> > **Q1:** In equation, (2) It would be clearer to define  **p** and *g*.
> >
> > **ANSWER:** We now define p and g in the revised manuscript, see below:
> >
> > **CHANGES TO MANUSCRIPT:
> >
> > We now include the following defining **p** and *g*:
> >
> > *“We evaluate the surface point distances $\tilde{d}$ and one-hot semantic labels $\mathbf{\tilde{h}}$ predicted by the decoder $g$ for query points surface points $\mathbf{p}$ and off-surface points $\mathbf{o}$ using following loss terms:”*
> >
> > **Q2:** If I understood correctly, to learn the VAE, points are sample in the surface and off-surface points are generated close to the surface points. It is sufficient to sample off-surface point only near the surface points to correctly learn the SDF ?
> >
> > **ANSWER:** Solely sampling points close to the surface might not be sufficient. Note that we normalize shapes to be within a [-1, 1] bounding box and sample off-surface points by adding zero mean Gaussian noise with a 0.3 standard deviation to on-surface points. This results in the off-surface points covering a wide region around the shape surface. It is important to have a wider distribution of points to ensure that the SDF gradients are consistent. Otherwise, because we do not supervise the SDF itself, zero level-set regions beyond the target surface could emerge, resulting in a more noisy SDF.
> >
> > **Q3:** In Figure 3, some real examples should be included to be compared with the generated vessel trees.
> >
> > **ANSWER:** We added real examples to the revised manuscript.
> >
> > **CHANGES TO MANUSCRIPT:**
> >
> > Updated Figure 3 to include real examples and updated the caption as follows:
> >
> > *“Real and synthetic semantic vessel trees from TopCoW. The \textbf{top} row shows real trees from. The \textbf{middle} row shows synthetic trees from the proposed method. The \textbf{bottom} row shows trees from \citet{kuipers2024generating} with failed reconstructions of the tree topology.”*
> >
> > **Q4:** The 1-NNA and COV metrics should be defined.
> >
> > **ANSWER:** We added the formal definitions of the 1-NNA and COV metrics to the Appendix.
> >
> > **CHANGES TO MANUSCRIPT:**
> >
> > We added Appendix section E that explain the 1-NNA and COV metrics, including their formal definitions:
> >
> > *"The 1-nearest neighbor accuracy (1-NNA) quantifies representativeness by measuring the similarity between real and synthetic shape distributions. It classifies each shape based on the set its nearest neighbor belongs to according to a distance metrics. In our case, we use Chamfer distance (CD) as a distance metric between point clouds. A 50% accuracy indicates no distinction between real and synthetic distributions, as half of the real and synthetic shapes are classified as synthetic and real, respectively. Following \cite{erkocc2023hyperdiffusion}, we formulate 1-NNA between a set of reference shapes $S_r$ and a set of generated shapes $S_g$ as*
> >
> > \begin{align} \text{1-NNA}(S_r, S_g) = \frac{1}{|S_r| + |S_g|}\sum_{X \in S_r}\mathbf{I}[N_X \in S_r] + \sum_{Y \in S_g}\mathbf{I}[N_Y \in S_g], \end{align}
> >
> > *where $|\cdot|$ denotes set-cardinality and $N_X$ the point cloud that is closest to $X$ from the union of the referene set and synthetic set:*
> >
> > \begin{align} N_X = \argmin_{K \in S_r \cup S_g}\text{CD}(X, K). \end{align}
> >
> > *Coverage (COV) assesses diversity by identifying the nearest real neighbor for each synthetic sample and calculating the ratio of unique real neighbors to total synthetic samples. We formulate COV between a set of reference shapes $S_r$ and a set of generated shapes $S_g$ as*
> >
> > \begin{align} \text{COV}(S_r, S_g) = \frac{1}{|S_r|} | \argmin_{X \in S_r}CD(XY)|Y \in S_g|. \end{align}"

---

### Author Rebuttal · Authors · 2025-03-07

**Rebuttal:**

We sincerely appreciate the reviewers’ constructive feedback, which has helped improve the clarity of our manuscript. A key point raised by all reviewers relates to benchmarking—specifically, the absence of direct baselines for comparison. Importantly, this is not a limitation of our method but rather a natural consequence of its novelty: existing vessel generation approaches impose constraints that make them inapplicable to our setting.

To address this, we have strengthened our discussion of related work to explicitly outline why prior methods are unsuitable for generating semantic cerebral vessel trees without topological restrictions. We have also introduced a dedicated "Evaluation Methodology" section to clarify our experimental choices and baseline selection. Additionally, we present expanded qualitative and quantitative comparisons, further demonstrating why previous methods fail to generalize to complex vessel structures.

With these revisions and our detailed responses to individual reviewer comments, we hope to have fully addressed any ambiguities in our original manuscript. We appreciate the reviewers' engagement with our work and look forward to further discussion.

We included the revised manuscript as a PDF with all changes highlighted in red.

**Supporting Material:**

/attachment/fc9b85967620ec512fd986c57adfeac8c2d37109.pdf

---

### Meta-Review · Area_Chair_J63h · 2025-03-19

**Recommendation:** Accept (Oral)
**Confidence:** 5

**Metareview:**

The reviewers are unanimous in their appreciation of this innovative work on synthetic vessel tree generation. The work is well-written and well-presented. The authors have actively engaged in the rebuttal period to clarify some outstanding issues. The paper would be an excellent fit for MIDL.